# Ultrafast hot-carrier dynamics in ultrathin monocrystalline gold

Can O. Karaman [1,3], Anton Yu. Bykov [2,3], Fatemeh Kiani[1], Giulia Tagliabue [1] ✉ & Anatoly V. Zayats [2] ✉

Applications in photodetection, photochemistry, and active metamaterials and metasurfaces require fundamental understanding of ultrafast nonthermal and thermal electron processes in metallic nanosystems. Significant progress has been recently achieved in synthesis and investigation of low-loss monocrystalline gold, opening up opportunities for its use in ultrathin nanophotonic architectures. Here, we reveal fundamental differences in hot-electron thermalisation dynamics between monocrystalline and polycrystalline ultrathin (down to 10 nm thickness) gold films. Comparison of weak and strong excitation regimes showcases a counterintuitive unique interplay between thermalised and non-thermalised electron dynamics in mesoscopic gold with the important influence of the X-point interband transitions on the intraband electron relaxation. We also experimentally demonstrate the effect of hot-electron transfer into a substrate and the substrate thermal properties on electron-electron and electron-phonon scattering in ultrathin films. The hot-electron injection efficiency from monocrystalline gold into $TiO_2$, approaching 9% is measured, close to the theoretical limit. These experimental and modelling results reveal the important role of crystallinity and interfaces on the microscopic electronic processes important in numerous applications.

Monocrystalline (MC) metals have recently emerged as attractive materials for a wide range of plasmonic applications, from sensing and imaging to information processing and energy harvesting[1–4]. In particular, because of the highly ordered crystal structure and lack of grain boundaries, they exhibit unique optical properties and allow for long-range plasmon coherence and reduced optical losses[5–8]. MC metals have in fact been shown to result in higher resonance quality factors, larger surface plasmon polariton propagation length, and superior electric field confinement compared to their polycrystalline (PC) counterparts[5,9,10]. Concurrently, advancements in the synthesis of MC metallic microflakes (MFs), in particular Au MFs, towards larger lateral sizes and higher aspect ratios, have enabled the study and exploitation of these outstanding physical properties in ultrathin (sub 15 nm) films, which are difficult to realise with PC metals[2,11].

Plasmonic-based charge transfer devices, such as photodetectors and photoelectrodes[12–14], could uniquely benefit from ultrathin MC metals. These systems indeed rely on the injection of photoexcited energetic charges (hot electrons and/or holes) from the metal into an adjacent semiconductor or adsorbed molecules[15–17]. Therefore, physical dimensions comparable to hot-carrier mean-free paths (sub 20 nm) and reduced scattering losses could favour ballistic hot-carrier transfer, significantly improving device performance. Additionally, well-defined crystal facets in MC metals would help clarify the interplay of catalytic properties and hot-carrier injection in plasmonic photoelectrodes[18]. In order to design and optimize the performance of hot-carrier devices, a detailed microscopic understanding of the carrier properties and thermalization dynamics in the material is essential.

Hot carriers initially generated in metal by light absorption have non-thermal energy distribution (which cannot be described by the

[1]Laboratory of Nanoscience for Energy Technologies (LNET), STI, École Polytechnique Fédérale de Lausanne, 1015 Lausanne, Switzerland. [2]Department of Physics and London Centre for Nanotechnology, King's College London, London WC2R 2LS, UK. [3]These authors contributed equally: Can O. Karaman, Anton Yu. Bykov. ✉e-mail: giulia.tagliabue@epfl.ch; a.zayats@kcl.ac.uk

Fermi-Dirac distribution) with energies that depend on the incident photon energy and the material band structure[19]. They are generated via intraband and interband absorption described by the dielectric function of the metal, $\epsilon = \epsilon_1 + i\epsilon_2$, and augmented by geometric effects in nanoscale structures[20]. Non-thermal carriers subsequently undergo a thermalization process that involves carrier-carrier scattering (typical carrier thermalization time $\tau_{th}$ of few hundreds of femtoseconds) and carrier-phonon relaxation ($\tau_{e-ph}$ of few picoseconds)[21–23], ultimately dissipating the photon energy into lattice heating. Ultrafast transient absorption spectroscopy studies of polycrystalline metal films and monocrystalline (colloidal) nanoparticles have shown that the substrate thermal conductivity as well as hot-carrier extraction strongly alter the overall thermalization dynamics[24–30]. Interestingly, nanoparticle-based studies have recently shown that the size and metal crystallinity can alter the electron-phonon coupling time[31–34]. Yet, because of limitations in the temporal resolution, there remains a lack of understanding of how the crystallinity affects the early thermalization stages (sub 100 fs), where electron-electron scattering is dominant. No study to date has investigated the hot-carrier dynamics in MC ultrathin films and transfer to semiconductors, both of which are essential for applications.

In this work, we study hot-carrier generation, relaxation, and transfer dynamics in ultrathin MC Au microflakes under different excitation conditions. We perform transient reflectance spectroscopy of these films with near-IR 8-fs pulses, providing insights into the hot-carrier thermalization dynamics on ultrashort timescales. In particular, we observe a decrease in electron scattering rate in MC gold, which suggests that grain boundaries may play a role in this process. Our results also demonstrate several features of hot-electron dynamics in the thermalized regime, such as the dynamic renormalization of the interband absorption peak at the X high symmetry point in the Brillouin zone as well as a strong contribution of hot-electron scattering on polar phonons in the substrate, caused by the electron spill-out and manifested by the dependence of the relaxation rate on the thermal

conductivity of the substrate and thickness of gold crystals. The latter behaviour is reversed for stronger optical excitation due to the increasingly higher energy stored in the hot-electron gas. Finally, we use the MC gold flakes as a platform for ultrafast hot-electron transfer into the adjacent semiconductor material (TiO$_2$) and demonstrate the injection efficiency as high as $\approx$ 9%, close to the theoretical limit, despite the Au surface being atomically flat. This finding is highly promising for hot-carrier transfer devices with monocrystalline metals, allowing high-quality factor surface plasmon resonances thanks to the low optical loss and their atomically flat surfaces. We also show the impact of the excitation regime (weak or strong perturbation), resulting in the suppression of the electron spill-out effect on the electron-phonon relaxation time as perturbation increases. The obtained results reveal the important effects of crystallinity on hot-carrier dynamics, providing opportunities for the development of plasmonic hot-carrier devices.

## Results

Ultra-thin MC Au MFs with pristine (no ligand) and atomically smooth (111) surfaces are grown on glass by a wet chemistry method[2]. The studied range of the flake thicknesses is 10 nm to 20 nm, as determined by atomic force microscopy (Supplementary Fig. 5a). The high aspect ratio of the flakes concurrently ensures lateral sizes >10 μm, suitable for optical spectroscopy. Separately, complementary 10-nm-thick, continuous PC Au films were prepared by sputtering. The hot-electron injection into TiO$_2$ was studied by transferring the chemically synthesized MC Au flakes onto a 40-nm-thick TiO$_2$ film, deposited on borosilicate glass (see Methods). Calculated reflection, absorption and transmission spectra of the 10-nm-thick MC Au (the dielectric function from ref. 35) and PC Au (the dielectric function from ref. 36) films on a SiO$_2$ substrate show small but distinguishable differences (Fig. 1c). In the spectral range of the pump beam used in the experiments, the mean absorption for PC films ($A = 0.071$) is slightly higher than that of MC films ($A = 0.066$).

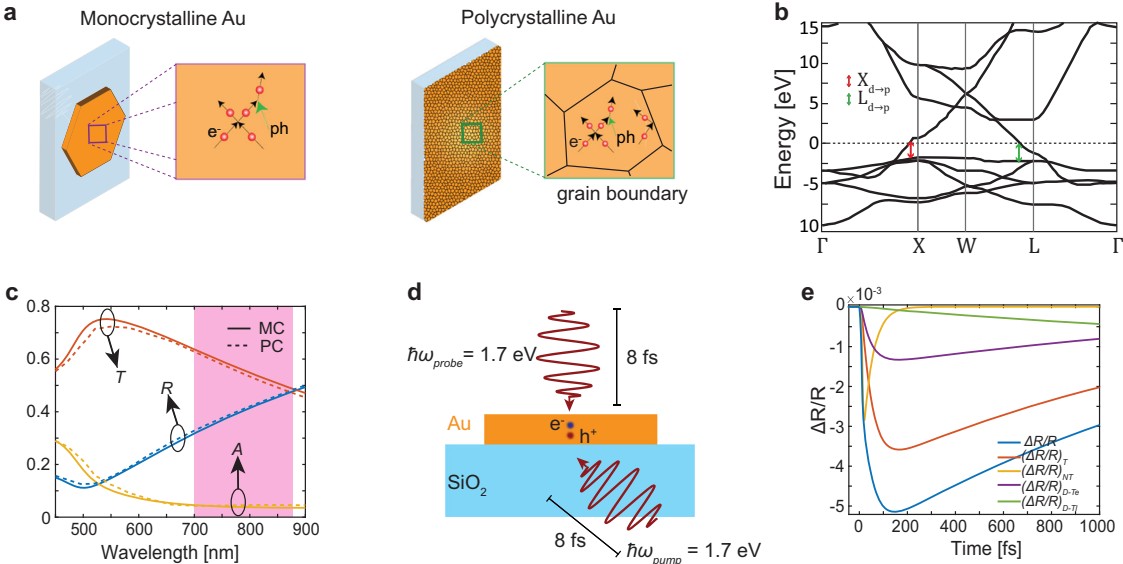

**Fig. 1 | Optical properties of monocrystalline gold. a** The schematic illustrations of monocrystalline and polycrystalline Au. **b** Schematic of the band diagram of Au. Red arrow indicates the transition from d-band to p-band near the X-symmetry point, the green arrow indicates the transition near the L-symmetry point. **c** The simulated reflection, absorption and transmission spectra of the 10-nm-thick MC Au (solid curves) and PC Au (dashed curves) on a SiO$_2$ substrate at normal incidence. The shaded area indicates a spectral range of the excitation. **d** Schematics of the degenerate pump-probe optical measurements with back surface pumping and front surface probing. **e** Simulated $\Delta R/R$ (blue curve) for a 10-nm-thick MC Au flakes

on a SiO$_2$ substrate at the laser fluence of 3.5 Jm$^{-2}$ and 8-fs-long pulses and its components $\Delta R_T/R$ (orange curve), $\Delta R_{NT}/R$ (yellow curve), $\Delta R_{D-Te}/R$ (violet curve), and $\Delta R_{D-Tl}/R$ (green curve). The nonthermalised, $\Delta R_{NT}/R$, and thermalized, $\Delta R_T/R$, contributions are simulated by considering in Eq. 1 only either $\Delta\epsilon_{NT}(\hbar\omega, t)$ or $\Delta\epsilon_T(\hbar\omega, t)$, respectively. The Drude contribution in the transient reflectivity is separated into two parts, which depend on an electron temperature, $\Delta R_{D-Te}/R$, or a lattice temperature, $\Delta R_{D-Tl}/R$, and computed by considering only either elevated $T_e$ or $T_l$, respectively. See Supplementary Note 1 for the details of the simulations.

Hot-carrier dynamics was investigated using transient reflectance measurements (Fig. 1d, see Methods). An 8 fs pump-pulse laser allows to measure the whole thermalization process, including at sub-100 fs time scales. Both pump and probe beams have a central photon energy of 1.7 eV. In Au, this corresponds mainly to the intraband hot-carrier excitation regime, which results in a near-uniform non-Fermi distribution of non-thermal carriers from the Fermi level up to the energy of the excitation photon[19,21,37,38]. At photon energies >1.8 eV, the interband transitions dominate the optical response and include transitions at X-point (≈ 1.8 eV, d-band to p-band transition) and at L-point (≈ 2.4 eV, d-band to p-band and ≈3.5 eV, p-band to s-band)(Fig. 1b, Supplementary Fig. 1).

The relaxation dynamics of the hot carrier ensemble can be well-described using a three-process model[19,39] that evaluates the time evolution of the energy density of the non-thermal hot carriers, $N_e$, the temperature of the thermalized electron gas, $T_e$, and the temperature of the lattice, $T_l$. This model, combined with a semiclassical theory of optical transitions in the solids, allows to establish a link between the non-equilibrium energy transfer and the measured changes in the dielectric function $\Delta\varepsilon(\hbar\omega, t)$ of the metal and, therefore, the measured reflectance signal (see Supplementary Note 1 for the details of the model)[19,23]. The changes in permittivity $\Delta\varepsilon$ can be split in three distinct contributions from (i) the non-thermal hot carriers ($\Delta\varepsilon_{NT}$), (ii) the thermalized electrons with the elevated temperature ($\Delta\varepsilon_T$), and (iii) the Drude damping modification ($\Delta\varepsilon_D$) as in ref. 19:

$$\Delta\varepsilon(\hbar\omega, t) = \Delta\varepsilon_{NT}(\hbar\omega, t) + \Delta\varepsilon_T(\hbar\omega, t) + \Delta\varepsilon_D(\hbar\omega, t) \qquad (1)$$

The importance of these contributions changes with the time after the excitation. The first two terms account for the transient change of the single-particle interband absorption peak due to the pump-induced change in the electron-hole occupancy and the electron temperature, respectively, while the last term accounts for the many-body free-electron response manifested by the change in the optical Drude damping. In the spectral range above the interband offset threshold, the main contribution is given by the $\Delta\varepsilon_T$, and a characteristic "slow" rise is observed in the optical constants as the internal equilibrium is being reached within the hot-electron gas (i.e., $\tau_{rise} \approx \tau_{th}$). In a highly non-equilibrium Fermi gas, which can be obtained at higher optical fluences the thermalisation time decreases due to stronger electron-electron scattering[40]. At longer wavelengths, on the other hand, $\Delta\varepsilon_{NT}$ dominates the response and exhibits an almost instantaneous $\tau_{rise}$, irrespective of the strength of the excitation, which depends on the average scattering rate of the non-thermal electrons[41,42] (Supplementary Fig. 2).

Figure 1e illustrates the contributions of the above-introduced excitation/relaxation processes to the transient reflection, $\Delta R/R$. Initially, the photon absorption results in a sudden change in the electron occupancy and creates a nonequilibrium, nonthermal electron distribution, which causes the rapid increase of the signal by $\Delta R_{NT}/R$. The nonthermalised electrons interact inelastically with each other and also with 'cold' electrons, not perturbed from their initial thermal equilibrium, resulting in the thermalized electron distribution after ~100s of fs. The contribution $\Delta R_T/R$, which accounts for the interband transitions, is associated with this new electron distribution, thermalised at a higher electron temperature. The thermalized electrons also impact the Drude response of the electron gas, resulting in $\Delta R_{D-Te}/R$. The thermalized electrons collide with the lattice of a metal, and their excess energy is transferred to phonons. Therefore, a lattice temperature increases over a few ps time, determined by the electron-phonon coupling constant of the system, $G$. As a consequence, the $\Delta R_{D-Tl}/R$ contribution is built up. The excited phonons interact with the room-temperature phonons of the lattice and the substrate phonons, and their energy is dissipated until the thermal equilibrium with the surroundings is established.

## Electron-electron relaxation

We first compare the behaviour of highly nonequilibrium hot electrons in MC and PC gold generated under strong excitation conditions (Fig. 2a). At low fluences, both samples exhibit exactly the same charge carrier dynamics. In particular, we observe a rapid rise, ($\tau_{rise} = 150$ fs $<< \tau_{th} \approx 500$ fs), dictated by the non-thermal electron dynamics $\Delta\varepsilon_{NT}$, as expected for the nonresonant (with respect to the interband transitions) probing conditions. However, when the fluence increases, two intriguing phenomena emerge: (i) counter-intuitively, $\tau_{rise}$ increases and, most significantly, (ii) $\tau_{rise}$ in the MC sample becomes longer than in the PC one, despite its lower absorption. This indicates different early-stage thermalization dynamics in MC and PC films.

Using $\varepsilon_{Au}(\omega)$ for PC Au[36] and the electron-electron scattering rate ($\gamma_{e-e}$, which has a quadratic dependence on electron energy[42,43]), we obtain an excellent agreement between the experimental and theoretical transient reflectance (Fig. 2c). Importantly, at higher fluences and hence higher $T_e$ of the thermal carriers, the joint density of states (JDOS) for the interband transitions at the X-point of the Brillouin zone broadens in energy so that the spectral overlap with the probe beam spectrum used in the measurements increases (Fig. 2d). In other words, when a high $T_e$ is established, the probe pulse can transiently access interband absorption at the X-point, becoming more sensitive to the thermal-carrier contribution ($\Delta\varepsilon_T$) similar to the resonant probing conditions of the interband transitions. The interplay between the evolution of $\Delta\varepsilon_T$ and $\Delta\varepsilon_{NL}$ then produces a "delayed" rise time, observed in the experiment. To confirm this conclusion, we compare the dependence of $\Delta\varepsilon_{NT}$ and $\Delta\varepsilon_T$ on the excitation fluence (Fig. 2b and Supplementary Fig. 2). At low fluences ($F = 0.4$ Jm$^{-2}$), the contribution from $\Delta\varepsilon_T$ is negligible, and the response is dominated by the fast $\Delta\varepsilon_{NT}$. However, at stronger excitation, the magnitude of $\Delta\varepsilon_T$ increases rapidly, reaching nearly 25% of the nonthermal contribution for a fluence of 5.7 Jm$^{-2}$. We conclude, therefore, that the observed experimental trends and their dependence on the excitation fluence showcase an overlooked unique interplay between equilibrium and non-equilibrium electron dynamics in mesoscopic gold. We can speculate that the stronger role of X-point in the dielectric function of MC gold is what allows us to more clearly observe it at higher fluences in the transient spectra (Fig. 2a). It is worth noting that we do not observe the decrease (increase) of $\tau_{th}$ ($\gamma_{e-e}$) that follows from the Fermi liquid theory[37,42] due to the predominantly nonresonant nature of the experiment, which is only partially sensitive to the non-equilibrium dynamics ($\tau_{rise} << \tau_{th}$).

For strongly perturbative excitation at higher fluence, the rise of $\Delta R/R$ in the PC Au film is faster than in the MC Au MF. Treating the average scattering rate of the nonthermalised electrons, $\overline{\gamma_e}$, as a free parameter of the dynamic three-process model (Supplementary Note 1) and noting the larger JDOS at the X-point for MC than PC Au (Fig. 2d), we can reproduce the measured electron dynamics in MC gold (Fig. 2e). The greater spectral weight of the interband transitions in MC Au means a stronger contribution of the slow $\Delta\varepsilon_T$ term to the measured dynamics with the obtained $\overline{\gamma_{e,MC}} = 18$ THz $< \overline{\gamma_{e,PC}} = 24$ THz. This difference in $\overline{\gamma_e}$ arises from the additional contributions from scattering on lattice defects and grain boundaries in PC Au, which are the additional energy loss channels for electrons[44–46]. The studied PC Au film has an average grain size of 50 nm (Supplementary Fig. 5) which is much smaller than the pump beam size (≈10 $\mu m$) and thus multiple grain boundaries contribute. We can, therefore, attribute the longer $\tau_{rise}$ in MC Au to the combination of (i) a decrease in $\overline{\gamma_e}$ and (ii) an increase of the contribution of the interband optical transitions at the X-point. The former also results in longer thermalization of electrons in MC Au.

To visualize the effect of hot-carrier extraction on the early-stage thermalization dynamics of hot carriers, we compare a 10-nm-thick Au MF on SiO$_2$ and n-doped TiO$_2$ substrates. At an Au/TiO$_2$ interface, a Schottky barrier of ~1.2 eV is formed[47–49] (Fig. 3a), enabling hot-electron

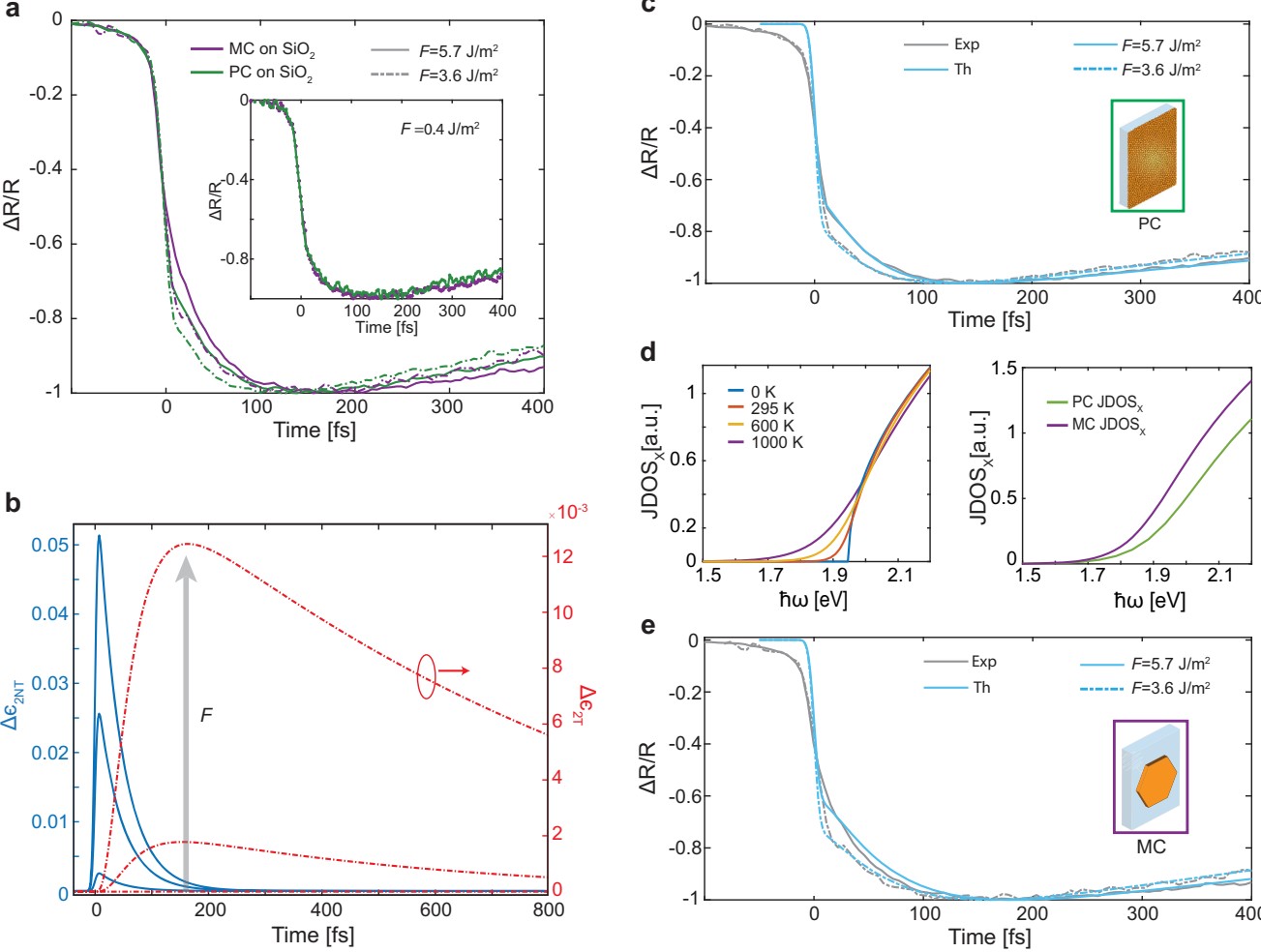

**Fig. 2 | Electron-electron scattering effects. a** Normalised transient reflection from a 10-nm-thick MC Au MF (violet curves) and a PC Au (green curves) on SiO$_2$ substrate at the pump fluence $F = 3.6$ Jm$^{-2}$ (dashed curves) and $F = 5.7$ Jm$^{-2}$ (solid curves). The inset shows the weak perturbation regime measurements with $F = 0.4$ Jm$^{-2}$. **b** Transient change in the imaginary part of the dielectric function of Au at $\hbar\omega = 1.7$ eV due to the nonthermalised $\Delta\epsilon_{NT}$ (blue curves) and thermalised $\Delta\epsilon_T$ (orange curves) electron distributions, simulated with the three process model.

**c** Simulated (cyan curves) and measured (gray curves) normalised $\Delta R/R$ from a 10-nm-thick PC Au film on SiO$_2$. **d** (left) JDOS at the X-symmetry point of the Brillouin zone of the MC Au at different electron temperatures and (right) JDOS at the X-symmetry point for MC (purple curves) and PC (green curves) Au. **e** Simulated (cyan curves) and measured (gray curves) normalised $\Delta R/R$ for a 10-nm-thick monocrystalline Au MF on SiO$_2$. In (**a**, **c**, **e**), dashed curves correspond to $F = 3.6$ Jm$^{-2}$ and solid curves to $F = 5.7$ Jm$^{-2}$.

separation across the metal/semiconductor interface. Under these conditions, regardless of the pump fluence, the rise of $\Delta R/R$ is slower than on SiO$_2$. This is related to the transfer of the non-equilibrium hot electrons with the energies exceeding the Schottky barrier from Au to TiO$_2$ in the first 100 fs, increasing the thermalisation time governed by the electrons closer to the Fermi level. In fact, $\gamma_{e-e}$ is quadratically proportional to the excited electron energy, and lower energy carriers have a longer lifetime (Supplementary Note 2)[23,42,43]. Since the remaining electrons are less energetic, they will live longer. This effect occurs at all levels of excitation as hot-carrier transfer is always present in Au/TiO$_2$ system. However, it is fundamentally different from the power-dependent phenomena discussed above, and it survives even in the regime of low perturbation excitation (Fig. 3b).

**Electron-phonon scattering and hot-electron transfer efficiency**
To complete the picture of the role of crystallinity and hot-electron extraction on the overall carrier thermalization dynamics, we analyze the temporal response of optical properties of MC and PC films governed by the electron-phonon scattering (Fig. 4b). Electron-phonon relaxation time $\tau_{e-ph}$ increases with the excitation fluence for all the samples since the free-electron heat capacity depends on thermalized

$T_e$, and, therefore, the relaxation is slower for higher initial values of $T_e$[27,33,50]. For the same excitation fluence, $\tau_{e-ph}$ is slightly shorter in the MC Au MFs compared to the PC Au thin films. We can estimate the electron-phonon coupling constant from the intersect of its dependence on the excitation fluence as $G_{MCAu} = (2.2 \pm 0.1) \times 10^{16}$ Wm$^{-3}$K$^{-1}$ and $G_{PCAu} = (2.0 \pm 0.1) \times 10^{16}$ Wm$^{-3}$K$^{-1}$, for MC and PC samples, respectively. Since both films are on the same substrate, the change in an electron-phonon coupling constant originates from the difference in crystallinity. In particular, grain boundaries can affect the phonon density of states and frequencies[51], therefore, impacting the rate of energy transfer from electrons to phonons[51], showcasing the importance of using robust single crystal platforms for the design of plasmonic hot-carrier devices.

A second practically important observation is a slower electron-phonon relaxation in MC Au on TiO$_2$ and a drastic difference in the electron-phonon coupling constants for the identical MC gold films on SiO$_2$ ($G_{MCAu} = (2.2 \pm 0.1) \times 10^{16}$ Wm$^{-3}$K$^{-1}$) and TiO$_2$ ($G_{MCAu/TiO_2} = (1.6 \pm 0.1) \times 10^{16}$ Wm$^{-3}$K$^{-1}$). Several studies have reported a major effect of the environment on $\tau_{e-ph}$ in Au NPs and Au thin films interfaced with dielectric materials[25,26,52–55]. In particular, it has been shown that $\tau_{e-ph}$ decreases in an environment with higher thermal conductivity which is

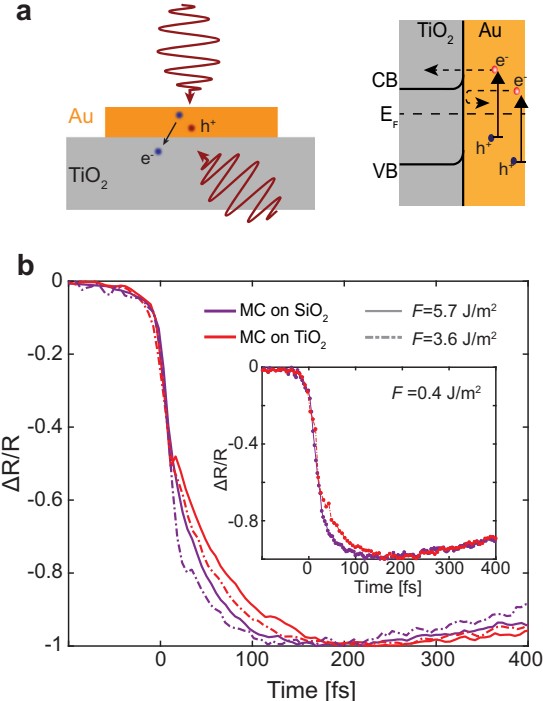

**Fig. 3 | Effect of hot-carrier transfer on the electron dynamics. a** The schematic illustration of hot-electron transfer from Au to TiO₂ (right) and the respective energy band diagram (left). **b** Normalised transient reflection measured for a 10-nm-thick monocrystalline Au MF on SiO₂ (purple curves) and on TiO₂ (red curves) at a pump fluence $F = 3.6$ Jm⁻² (dashed curves) and $F = 5.7$ Jm⁻² (solid curves). Inset shows the low perturbation $\Delta R/R$ measurement at $F = 0.4$ Jm⁻².

likely to be related to the scattering of Au electrons on the phonons in the substrate[53]. Therefore, we expect smaller $G_{MCAu/TiO_2}$ to partially originate from the lower thermal conductivity of thin TiO₂ film compared to bulk SiO₂[56]. Hot-electron injection is also expected to modify $\tau_{e-ph}$. Since the most energetic electrons will be transferred, the thermalised electron gas temperature will increase less in the presence of hot-carrier transfer. Therefore, one could also expect a decrease in $\tau_{e-ph}$ in the case of a MC Au MF on TiO₂, as it depends on the electron temperature.

To disentangle the two contributions, we consider the relationship between $\tau_{e-ph}$ and the hot-electron injection probability, $P_i$[33,50]:

$$\tau_{e-ph} \approx \frac{\gamma T_l}{G_{MCAu/TiO_2}} + \frac{(1 - P_i)U}{2G_{MCAu/TiO_2} T_l} \quad (2)$$

where $\gamma = 66$ Jm⁻³K⁻² is the electron heat capacity for Au, and $U$ is the initial energy density absorbed by the electrons $U = A \times F/L$, where $A$ and $L$ are the absorbance of the film and its thickness, respectively. Using $P_i$ as a fitting parameter, the dependence of the electron-phonon relaxation time on the excitation fluence can be fitted to the experimental data with $P_i = 0.09 \pm 0.03$ (Fig. 4b). On the other hand, the theoretical hot-electron injection probability from a monocrystalline Au MF into TiO₂ can be evaluated, assuming $\Phi_b = 1.2$ eV and $\hbar\omega = 1.7$ eV, as $P_i = 0.1$ (see Supplementary Note 2 for the details of the calculations). Surprisingly, there is an excellent agreement between the hot-electron injection probability obtained from the theoretical estimates under fully-relaxed momentum conservation and the one extracted from the experimental $\tau_{e-ph}$ dependence.

Several studies demonstrated the increase in the hot-electron transfer efficiency due to electron-surface-defect scattering which causes the electron momentum randomization if the surface is rough or due to a grain structure of PC metals[57,58]. However, our results

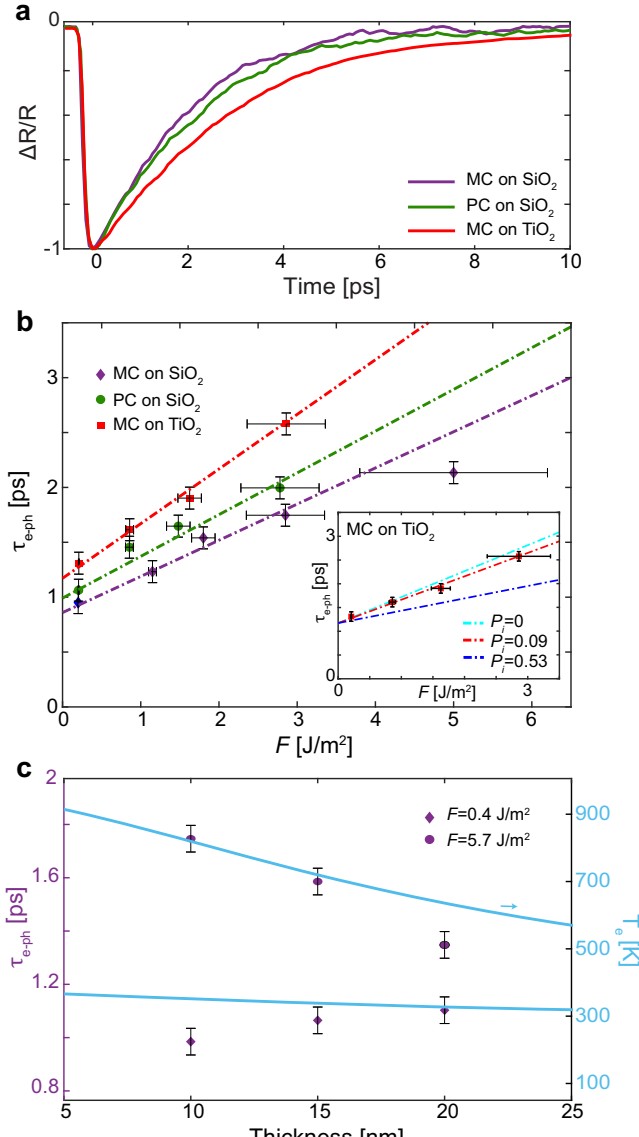

**Fig. 4 | Electron-phonon scattering. a** Measured normalised transient reflection from a 10-nm-thick MC Au MF (violet curve) and a PC Au film (green curve) on a SiO₂ substrate, and a MC Au MF on a TiO₂ substrate (red curve) at the excitation fluence $F = 5.7$ Jm⁻². **b** Measured dependence of $\tau_{e-ph}$ of a 10-nm-thick MC Au MF on SiO₂ (violet diamonds), on TiO₂ (red squares), and a PC Au film on SiO₂ (green dots) on the excitation fluence. The dotted curves are simulated with Eq. (2) for $G_{MCAu/TiO_2} = 1.6 \times 10^{16}$ Wm⁻³K⁻¹ (red), $G_{PCAu/SiO_2} = 2 \times 10^{16}$ Wm⁻³K⁻¹ (green), and $G_{MCAu/SiO_2} = 2.2 \times 10^{16}$ Wm⁻³K⁻¹ (purple). Inset shows the comparison of the fluence dependence of $\tau_{e-ph}$ of a 10-nm-thick MC Au MF on TiO₂ with the simulations for different values of $P_i$. **c** Thickness dependence of measured $\tau_{e-ph}$ for MC Au MFs on SiO₂ in strong (dots), and weak (diamonds) perturbation regimes. Solid curves are the calculated electron temperatures $T_e$ achieved in strong (upper) and weak (lower) perturbation regimes. Error bars in (**b**, **c**) represent standard deviation.

suggests that, despite their atomically smooth surfaces, MC Au MFs provide good hot-electron extraction efficiency (≈10%). This agrees well with the reduced hot-electron scattering rate measured above, as it would favor the ballistic extraction of the highly energetic electrons. Additionally, for the thicknesses lower than the electron mean-free-path, which is electron-energy-dependent and >30 nm at 1.7 eV photon energy[58], quasi-elastic electron-phonon scattering can efficiently redirect the momentum of hot electrons, which increases the transfer efficiency even in the absence of other momentum relaxing

mechanisms, such as lattice defects, grain boundaries, and rough surface scattering. Therefore, we assume the relaxation of the momentum conservation during hot-electron transfer, which shows good agreement with the measured efficiency. It should be noted that an ideal theoretical limit of the injection probability in the considered system can be obtained assuming $\Phi_b = 0$ eV, zero-thickness of Au film, and angular momentum for all the hot electrons directed towards the film interface to be $P_{i,max} = 0.53$.

Finally, we compare the electron-phonon relaxation times $\tau_{e-ph}$ for monocrystalline Au MF with different thicknesses (Fig. 4c). In a weak perturbation regime ($F = 0.4$ Jm$^{-2}$), the increase of the electron temperature $T_e$ is simulated to be stronger for the thinner films. However, interestingly, $\tau_{e-ph}$ decreases as the thickness decreases. While the contribution of surface acoustic[59] or electronic[60] states on highly oriented (111) Au surfaces may become relevant with the increase of a surface-to-bulk ratio, multiple previous theoretical and experimental works have failed to observe their role in hot-electron dynamics on surfaces even for much higher surface-to-bulk ratios than in the studied MC Au films[60–62]. Moreover, since the experimental studies are performed here at ambient conditions, the surface states are likely to be passivated by molecular contaminants[63]. Therefore, we attribute the decrease of $\tau_{e-ph}$ with the increase in thickness to the increased contribution of the electron spill-out effects, which reduces the average electron density for the thinner films[64]. This leads to the reduced screening and stronger interaction between the electrons and the ionic lattice[40,65]. The opposite behavior is observed at higher fluences in the strong perturbation regimes: $\tau_{e-ph}$ increases as the thickness decreases. The reason is that the non-negligible increase in $T_e$ (Fig. 4c), overpowers the contribution of the electron spill-out effect on $\tau_{e-ph}$: as the thickness decreases, the hot-electron temperature $T_e$ significantly increases in a high perturbation regime, and its increase is stronger for thinner films because of the smaller volume where the energy is absorbed, resulting in longer electron-phonon relaxation.

## Discussion

We interrogated temporal dynamics of hot carriers in ultrathin monocrystalline and polycrystalline Au films on passive and active substrates. Our findings revealed that a relative contribution of thermalised and nonthermalised electrons to transient optical properties depends on the crystallinity of a gold film. We found longer electron thermalization in monocrystalline Au due to a decrease in $\overline{\gamma_e}$, which was caused by the absence of grain boundaries and lattice defects. Contrary to general perception, in the near-IR spectral range, the interband transitions in gold at the X-point of the Brillouin zone significantly affect hot-carrier dynamics and must be taken into account in monocrystalline gold, especially in a strong perturbation regime at the high excitation fluences. This results in the increase of the contribution of long-lived thermalized electrons to the optical response and, therefore, its slower dynamics. We also showed that the presence of hot-electron transfer from Au to TiO$_2$ suppresses the nonthermal electron contribution to the optical properties and leads to much longer electron-phonon relaxation on a thermally low conductive substrate. We demonstrated that the perturbation regime highly affects the thickness dependence of the electron-phonon relaxation time $\tau_{e-ph}$. While $\tau_{e-ph}$ decreases as thickness decreases in a low perturbation regime owing to the electron spill-out effect, it increases in a high perturbation regime because of the non-negligible increase in $\Delta T_e$ for smaller thicknesses. Our findings also revealed that hot-electron injection efficiency in TiO$_2$ is as high as ≈9%, which agrees with the estimates under the assumption of momentum relaxation, even though the Au surface is atomically flat. This result indicates a potential for using monocrystalline metals in hot-carrier transfer devices[47,66,67], as it supports high-quality factor plasmonic resonances. Overall, the results contribute to a deeper understanding of the non-equilibrium carrier dynamics in monocrystalline metals.

## Methods
### Sample preparation
High-aspect ratio ultrathin (10–25 nm) monocrystalline Au MFs are fabricated by the procedure described in ref. 2. The on-substrate growth method results in MC Au MFs, which are directly nucleated and grown on the glass substrate surface with no organic or halide ligands present at the Au-glass interface; i.e., the bottom Au(111)/glass interface is pristine. The substrate is 180-μm-thick borosilicate glass. MFs have atomically smooth (Supplementary Fig. 5) and well-defined (111) crystallographic surface and face-centered cubic crystal structure (see ref. 2 for the details of the crystallographic characterization). After synthesis, a <2-nm-thick organic-halide adlayer is present on the top gold surface. The physically adsorbed organic-halide residue is easily removed with a simple cleaning procedure (several rinses with ethanol and DI water followed by drying with nitrogen gas) as it is not strongly bonded to the metal surface. The RMS roughness of the MF surface is measured to be ≈250 pm. To study hot-electron transfer, chemically synthesized flakes are transferred onto 40-nm-thick TiO$_2$ films on borosilicate glass by the PMMA transfer method[68], followed by the exposure to an oxygen plasma (4 min, 500 W) to remove any PMMA residue left from the transfer step. After the transfer, the pristine interface of the Au flakes (which was a Au/SiO$_2$ interface) is in contact with TiO$_2$. The detailed characterization of the MC Au-TiO$_2$ interface can be found in ref. 69. To compare the effect of crystallinity on hot-carrier dynamics, 10-nm-thick Au films were fabricated on the same type of substrates by sputtering. AFM imaging verifies that these films are continuous with the RMS surface roughness around 1.5 nm (Supplementary Fig. 5). The average grain sizes of the polycrystalline Au films are measured to be around 50 nm.

### Degenerate pump-probe measurements
Transient optical measurements were conducted using a femtosecond laser (Laser Quantum Venteon) and a degenerate pump-probe setup (Supplementary Fig. 5). The laser produces ≈8 fs pulses with a repetition rate of 80 MHz and an average power of 0.5 W. The spectrum of such a short laser pulse covers a range from 650 to 950 nm (Supplementary Fig. 5). The pump and probe beams were cross-polarized in order to prevent coherent interactions during the measurements. The measurements were performed using a lock-in amplifier and a modulated pump at 1 kHz frequency, which allowed for accurate measurements with a precision of $10^{-6}$–$10^{-7}$. The optical path included a dispersion control system (Sphere Photonics D-Scan) in order to control the dispersion of the ultrashort pulses. To assure consistency, $\Delta R/R$ traces of each sample were measured several times over a few different days.

## Data availability
All the data supporting the findings of this study are presented in the Results section and Supplementary Information and available from the corresponding authors upon reasonable request.

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

## Acknowledgements
This work was supported in part by the SNSF Eccellenza Grant #PCEGP2_194181, the ERC iCOMM project (789340), and the UK EPSRC project EP/W017075/1.

## Author contributions
G.T. and A.V.Z. conceived the project. F.K. fabricated the samples. C.O.K. and A.Yu.B. performed the experiments. C.O.K. performed the simulations. All the authors analysed the results and wrote the paper. G.T. and A.V.Z. supervised all aspects of the project.

## Competing interests
The authors declare no competing interests.
