## [Peer Review File · Nature Communications]

Reviewers' comments:

Reviewer #1 (Remarks to the Author):

The manuscript authorized by Anatoly et al is insightful and elucidates the hot-electron dynamics in monocrystalline and polycrystalline ultrathin gold. By utilizing transient reflectance spectroscopy with near-IR 8-fs pulses, the authors comprehensively clarified the hot-electron dynamics in gold films by analyzing the scattering rate, excitation regimes and transfer kinetics. The results reveal the important role of crystallinity and interfaces in microscopic electronic processes. This study is very systematic and provides guidance for realizing high-performance plasmonic hot-carrier devices. Therefore, the referee suggests the publication of this manuscript after the following revisions:

1. Figure 1c only illustrates the simulated reflection, absorption and transmission spectra. Whether the scattering of incident light should be considered, especially for polycrystalline gold film?
2. To make it more intuitive, the physical meanings of different stages in transient reflection spectra need to be elucidated. In addition, the components of the $\Delta R/R$ signal corresponding to the physical processes of electron-electron scattering, electron-phonon scattering and phonon-phonon scattering need to be illustrated.
3. Commonly, the width of the laser pulse should be excluded when fitting and extracting the relaxation times from ultrafast spectroscopy. In addition, the details of curve fitting should be illustrated in Supporting Information.
4. It was reported that grain boundaries can reflect charge carriers, which can alter the momentum direction and injection efficiency of hot electrons (ACS Photonics 2018, 5, 3613–3620). The related impact of grain boundaries in this model also needs to be discussed in detail.
5. The theoretical efficiency of hot-electron injection is widely described with Fowler's law. What's the difference between the calculation method in this paper and Fowler theory?
6. Transient reflection spectra and hot-electron dynamics of polycrystalline gold films with different grain sizes should be studied since the finite mean free path of carriers may affect their subsequent scattering and ballistic transport characteristics.
7. More crystal analysis should be performed for characterizing the crystallinity and microscopic morphology of gold film, such as TEM and XRD.
8. Is there any other lattice defects introduced in gold film during sputtering? Please discuss the impacts of defects in hot-electron dynamics.
9. The curve data in Figure 2 c and e are desultory, and not smooth enough.

Reviewer #2 (Remarks to the Author):

In this work, the thermomodulation response and interfacial transport dynamics in monocrystalline and polycrystalline Au thin films are investigated by transient reflection measurement and the three-process model based on Boltzmann-heat equations. The author shows that the band renormalization is induced by hot carriers, and that the boundary crystallinity affects the relative contribution of thermalized and nonthermalized electrons to the transient optical properties. The comparison of SiO₂ and TiO₂ substrates shows that there is hot-electron transfer from metal to substrate. The experimental and theoretical studies of non-equilibrium electron and phonon dynamics related to metal band structure, collision rates, and pump fluence have already been extensively reported in, such as ACS Photonics, 2016, 3(9):1637-1646; Light: Science & Applications, 2019, 8(1): 89; Phys. Rev. B 51, 11433 (1995); Phys. Rev. B 61, 16956 (2000); Nat. Comm. 5, 5788 (2014); Eur. Phys. J.D. 45, 369 (2007). Compared with previous works, the author mainly studied the effect of the dynamic band renormalization and electron scattering in non-equilibrium state induced by high fluence excitation. The originality and novelty it is not convincing of a significant advancement. I don't think

that the paper meets the standards of importance and novelty required by Nature Communications, and recommend to transfer the work to more focused journal.

Specific comments can be found below.

1、 Fig. 2a shows that monocrystalline Au thin films consistently exhibit a longer rising time than polycrystalline Au films with high pump fluence. Nanoparticles with different diameters show similar hot electron behavior and electron-phonon interaction, which is caused by the weak coupling between the free electron gas and the surface electron state, and the coupling between the hot electrons and the surface accounts for less than 10% of the total electron energy losses for these particles. (The Journal of Chemical Physics 112.13 (2000): 5942-5947; ACS Photonics, 2016, 3(9): 1637-1646.) For the Au nanofilms with enhanced influence of surface electronic states, how about the hot carrier dynamics behavior in monocrystalline Au films with varying thicknesses?

2、 When discussing the effect of thickness and substrate on the carrier dynamics, the author neglects the effect of surface electronic states of Au thin films. I am not convinced. As the thickness of the single crystal Au thin film gradually decreases, the body surface area ratio of the sample changes, and the influence of the electronic state on the surface of the sample increases. The surface modification effect of the substrate on the film should be considered.

3、 The Supplementary Fig. 2 (page 5, line 11) and relevant description of Fig. 2b is missing or mislabeling. The relevant parameters of rise under different conditions are not listed in the text and the supplementary materials.

Reviewer #3 (Remarks to the Author):

In this manuscript, the thermalization of the hot carriers and interfacial transport dynamics in monocrystalline and polycrystalline Au thin films with TiO₂&SiO₂ substrate are investigated by transient reflection measurement. The work claimed they have revealed the fundamental significantly work for the hot electron transfer between Au thin film and the TiO₂. However, I could not recommend its publication in Nature Communications for the reason below.

1. The significance of this finding is open to debate, its indeed that increasing the hot electron transfer rate from Au to TiO₂ is important, however, the physical origin and model that the non-thermalized carriers in Au film that modulated by the grain boundaries could affect the hot electron transfer rate in the Au/TiO₂ is unclear and should be reconsidered.

2. The comparison between the monocrystalline and polycrystalline Au thin films is unconvincing, for example, the band structure of the Schottky contact of the Au/SiO₂ Au/TiO₂ interface and the roughness level at the real space could not be as simple as the author modeled, which could inevitably affect the relaxation process. (All the parameters in Fig. S4 is hard to tell). They have not presented the grain boundary or other crystalline parameters neither for the pump beam spot size compared with the Au film, why could they confirm the relaxation contribution of the polycrystalline Au?

3. As shown in Fig. 4b, the e-ph coupling process for low pump fluence increased vividly with the thickness of the sample, which is in discrepancy with the Te model. The issue is that the grain size of the Au thin film is not presented or remain unchanged with the thickness therefore the conclusion of the increasing electron transfer should be questionable.

Reviewer #1 (Remarks to the Author):

The manuscript is insightful and elucidates the hot-electron dynamics in monocrystalline and polycrystalline ultrathin gold. By utilizing transient reflectance spectroscopy with near-IR 8-fs pulses, the authors comprehensively clarified the hot-electron dynamics in gold films by analyzing the scattering rate, excitation regimes and transfer kinetics. The results reveal the important role of crystallinity and interfaces in microscopic electronic processes. This study is very systematic and provides guidance for realizing high-performance plasmonic hot-carrier devices. Therefore, the referee suggests the publication of this manuscript after the following revisions:

1. Figure 1c only illustrates the simulated reflection, absorption and transmission spectra. Whether the scattering of incident light should be considered, especially for polycrystalline gold film?

Answer. We thank the reviewer for this comment. Indeed, the scattering contribution would be important in the case of a rough surface of the films. For single crystal gold, the surface is atomically flat (Figure R1a; also see Ref. 2 in the manuscript: Kiani, F. & Tagliabue, G. High Aspect Ratio Au Microflakes via Gap-Assisted Synthesis. Chemistry of Materials 34, 1278–1288 (2022) for the detailed characterization of the monocrystalline Au flakes identically prepared by the co-authors of this paper), and this effect can be neglected. In turn, based on the AFM images of the polycrystalline Au thin film (Figure R1b), the root mean squared surface roughness is about 1.5 nm. For this roughness level, no significant scattering from the continuous Au thin film is expected in the visible and near-infrared spectral range. Therefore, the simulated reflection, absorption, and transmission spectra fully represent the optical properties of the Au thin films.

We also would like to point out that in the experiments, additional scattering of light may arise due to the microflake boundaries if the beam is larger than a studied microflake. In our case, the probe beam, which is about 1 μm in diameter, is always centred on the flake, which has the dimensions of 30x30 μm^2 . Therefore, we do not observe any scattering from the edges of the flake during the experiments.

Figure R1. AFM height images of (a) monocrystalline Au MF, and (b) polycrystalline Au thin film used in the study.

Revision. We have the above discussion and the Figure R1 in in Supporting Information.

2. To make it more intuitive, the physical meanings of different stages in transient reflection spectra need to be elucidated. In addition, the components of the $\Delta R/R$ signal corresponding to the physical processes of electron-electron scattering, electron-phonon scattering and phonon-phonon scattering need to be illustrated.

Answer. We are grateful for this comment which helps to improve clarity of the presentation. To be more illustrative about the physical meanings of the different stages of the $\Delta R/R$ signal, we replaced Figure 1c of the original manuscript with a new Figure R2. Using the model, we introduced in the paper, we presented $\Delta R/R$ as its components, which are the contributions of nonthermal electrons, $(\Delta R/R)_{NT}$, thermalized electrons, $(\Delta R/R)_T$, and Drude contributions of electrons and phonons, $(\Delta R/R)_D$. First, absorption of a photon creates a sudden change in the electron occupancy and a nonequilibrium (nonthermal) electron distribution which causes the rapid increase of the signal $(\Delta R/R)_{NT}$. The nonthermalized electrons interact elastically between each other and with the “cold” electrons which after the excitation initially remained in the thermal equilibrium, resulting in the thermalized hot electron distribution after 100s of fs. The $(\Delta R/R)_T$ change is associated with this new hot-electron distribution thermalised at a higher electron temperature and accounts for the interband transitions. These thermalized electrons also have an impact on the Drude response of Au and result in a $(\Delta R/R)_{D-Te}$ contribution. The thermalized electrons collide with the lattice of a metal, and their excess energy is gradually transferred to phonons. Therefore, the lattice temperature increases over a few ps time, determined by the electron-phonon coupling constant of the system, G . As a consequence, $(\Delta R/R)_{D-Tl}$ contribution is built up. In meantime, these excited phonons interact with the room temperature phonons of the lattice and also the substrate phonons in longer time scales, and their energy is dissipated until the thermal equilibrium with the surroundings is established.

Figure R.2: The simulated $\Delta R/R$ (blue curve) for a 10-nm-thick monocrystalline Au film on a SiO_2 substrate for 8-fs pulses with a fluence of 3.5 Jm^{-2} and its components $(\Delta R/R)_T$ (orange curve), $(\Delta R/R)_{NT}$ (yellow curve), $(\Delta R/R)_{D-Te}$ (violet curve), and $(\Delta R/R)_{D-Tl}$ (green curve).

Revision. We replaced Figure 1e of the original manuscript with Figure R2 and added the above discussion on page 4.

3. Commonly, the width of the laser pulse should be excluded when fitting and extracting the relaxation times from ultrafast spectroscopy. In addition, the details of curve fitting should be illustrated in Supporting Information.

Answer. This is an important comment. The laser pulse length in our experiments is 8 fs which is much smaller than the reported relaxation times in the paper (roughly $\tau_{rise} \approx 250$ fs and $\tau_{e-ph} \approx 2$ ps. In this case, when we deconvoluted the measured $\Delta R/R$ with the 8-fs long gaussian shaped pulse, we did not observe any change in the measured lifetimes since the laser pulses are much shorter compared to the changes in the reflectivity. We always show the measured data of the fast rise of $\Delta R/R$ with the explanations of the fitting procedure to prevent any possible misunderstanding. Figure R3 shows one of the examples of exponential fitting to find τ_{e-ph} . We only used a single exponential fitting which is only needed for finding τ_{e-ph} by considering τ_{e-ph} as fitting parameter. The single exponential is fitted from the absolute maximum of $\Delta R/R$. In this specific example, the measured $\tau_{e-ph} \approx 1.27$ ps.

Figure R3. (a) The measured $\Delta R/R$ of a 15-nm-thick Au flake (yellow) and the best fitted single exponential (red).

Revision. We have added Section S3 in Supporting Information to clarify the fitting procedure.

4. It was reported that grain boundaries can reflect charge carriers, which can alter the momentum direction and injection efficiency of hot electrons (ACS Photonics 2018, 5, 3613–3620). The related impact of grain boundaries in this model also needs to be discussed in detail.

Answer. We are grateful for this important comment. Indeed, the scattering on grain boundaries may influence the injection efficiency in polycrystalline films as was previously reported and investigated. Since the main aim of the study is revealing the unexplored features of the monocrystalline Au in terms of hot carrier generation and relaxation, we did not focus on a charge transfer measurement with well-studied polycrystalline Au. For monocrystalline Au, as there is no scattering on grain boundaries, we assume the momentum direction of the photo-excited hot electrons is isotropic in space. The main conclusion from the model is that the measured collection efficiency of monocrystalline Au is of the same order of magnitude as the theoretical calculations in the simple model under momentum relaxation assumption at the interface. This result is important because it is known that the roughened surfaces are favourable for high carrier transfer efficiency due to randomizing the momentum

of the electrons by electron-surface collisions. But the agreement between the measurements and the ideal theoretical calculations implies that it is still possible to have high transfer efficiency at atomically smooth surfaces, and we think that it is because of the efficient randomization of the momentum of the reflected electrons from the surface by electron phonon scatterings in very thin films. The interface models like ours have been successfully used in the literature for studying injection into TiO₂ from plasmonic gold nanoparticles [1].

[1] Ratchford, D. C., Dunkelberger, A. D., Vurgaftman, I., Owrutsky, J. C., & Pehrsson, P. E. (2017). Quantification of efficient plasmonic hot-electron injection in gold nanoparticle–tio2 films. *Nano Letters*, 17(10), 6047–6055; Andrew J. Leenheer, Prineha Narang, Nathan S. Lewis, Harry A. Atwater; Solar energy conversion via hot electron internal photoemission in metallic nanostructures: Efficiency estimates. *J. Appl. Phys.* 7 April 2014; 115 (13): 134301].

Revision. We have added in the revised manuscript a concise discussion about the role of grain boundaries in electron injection on page 8.

5. The theoretical efficiency of hot-electron injection is widely described with Fowler’s law. What’s the difference between the calculation method in this paper and Fowler theory?

Answer. As Reviewer noted, the Fowler’s law is widely used to describe the photoemission from the emitter to the collector. The method we used in the article has the exactly same assumptions as the Fowler’s law [1] but with additional consideration of the film thickness (the original Fowler’s law is derived for bulk systems). The semiclassical model of internal photoemission, the model used in our paper, involves three steps: hot-electron excitation, hot-electron transport to the interfacial barrier, and hot-electron emission over the energetic barrier from the emitter material into the collector material. Although the actual processes of light absorption and excitation of the collective electron cloud are quantum-mechanical phenomena, we assume herein that after light absorption, the “hot electron” behaves as a quasiparticle whose transport can be described semiclassically within a free-electron-like band structure. The one of the main criteria in the Fowler’s law is for electron emission to occur, the kinetic energy perpendicular to the barrier must exceed the height of the barrier which ends up with a momentum conservation cone which is determined by the Fermi level, the hot-electron energy and the barrier height. Both models approximate the same expression for the case of bulk materials [2]. Yet, it has been shown that for nanostructures, this simple momentum conservation description does not hold and efficiencies higher than the Fowler’s law estimation have been reported both experimentally and numerically [2-4]. That is because the Fowler’s law only accounts for the single interaction with the metal-semiconductor interface and disregards the carrier interactions. Especially, when the thickness is lower than the hot-electron mean-free-path, the quasi-elastic scattering between electrons and phonons can efficiently randomize the momentum of the hot electrons that increase the transfer efficiency by increasing the probability of the momentum matching. Therefore, in our model, we assume the relaxation of the momentum conservation during the hot electron transfer which later shows the agreement with the measured efficiency value.

[1] V. V. Afanas’ev, *Internal Photoemission Spectroscopy: Principles and Applications* (Elsevier, Amsterdam, 2008).

[2] Andrew J. Leenheer, Prineha Narang, Nathan S. Lewis, Harry A. Atwater; Solar energy conversion via hot electron internal photoemission in metallic nanostructures: Efficiency estimates. *J. Appl. Phys.* 7 April 2014; 115 (13): 134301.

[3] Blandre, E., Jalias, D., Petrov, A. Yu., & Eich, M. (2018). Limit of efficiency of generation of hot electrons in metals and their injection inside a semiconductor using a semiclassical approach. *ACS Photonics*, 5(9), 3613–3620.

[4] Ratchford, D. C., Dunkelberger, A. D., Vurgaftman, I., Owrutsky, J. C., & Pehrsson, P. E. (2017). Quantification of efficient plasmonic hot-electron injection in gold nanoparticle–tio2 films. *Nano Letters*, 17(10), 6047–6055.

Revision. Based on the above discussion, we added a brief clarification in Section 2 of Supplementary Information.

6. Transient reflection spectra and hot-electron dynamics of polycrystalline gold films with different grain sizes should be studied since the finite mean free path of carriers may affect their subsequent scattering and ballistic transport characteristics.

Answer. We want to thank Reviewer for this comment and we agree that the grain size may affect the relaxation of the carriers, as was widely investigated previously [1,2]. The main aim of this study is revealing the hot-carrier dynamics and the transfer properties in atomically smooth monocrystalline Au which is the most ideal platform for the theoretical and practical studies and has not been used extensively so far. Therefore, most of our effort in this study goes to investigating monocrystalline Au both experimentally and theoretically and the time-resolved measurements of polycrystalline Au are performed for reference. For several aspects, such as hot-electron transport and the thickness dependent electron-phonon relaxation parts, we exclusively focus on monocrystalline Au because the electron momentum redistribution thanks to the carrier scatterings plays a major role in efficiency in absence of grain boundaries and uniform thickness all over the Au flake makes the effects of surfaces in carrier relaxation more pronounced as compared to the polycrystalline Au.

[1] Chawla, J. S., Gstrein, F., O'Brien, K. P., Clarke, J. S. & Gall, D. Electron scattering at surfaces and grain boundaries in Cu thin films and Wires. *Physical Review B* **84**, (2011).

[2] Feldman, B., Park, S., Haverty, M., Shankar, S. and Dunham, S.T. (2010), Simulation of grain boundary effects on electronic transport in metals, and detailed causes of scattering. *phys. stat. sol. (b)*, 247: 1791-1796. <https://doi.org/10.1002/pssb.201046133>

Revision. We checked that the revised text underlines the objectives of this study and that previous studies of polycrystalline Au are appropriately referenced.

7. More crystal analysis should be performed for characterizing the crystallinity and microscopic morphology of gold film, such as TEM and XRD.

Answer. We completely agree that morphology of the films is of utmost importance for this study. To characterize the morphology and the grain sizes, we used AFM which can be seen in Figure R1, which gives us very useful information such as the grain sizes and the surface roughness of polycrystalline Au. Therefore, we think that the AFM characterization should be enough and TEM analysis does not provide additional information in the scope of the paper. At the same time, the detailed characterization of the monocrystalline Au flakes identically prepared by the co-authors of this paper has been published and referenced in the manuscript (Ref. 2): Kiani, F. & Tagliabue, G. High Aspect Ratio Au Microflakes via Gap-Assisted Synthesis. *Chemistry of Materials* **34**, 1278–1288 (2022) - Ref. 2 in the manuscript. We are, therefore, reluctant to plagiarise ourselves with repeated presentation of the detailed morphological analysis.

8. Is there any other lattice defects introduced in gold film during sputtering? Please discuss the impacts of defects in hot-electron dynamics.

Answer. In addition to our answer to the comment 6, we want to point out that the main purpose of the study is revealing the unexplored capabilities of the monocrystalline Au rather than focusing the effect of crystal defects in polycrystalline Au, which was previously studied. We completely agree that the lattice defects might play a role in hot carrier relaxation in the exact same way that the grain boundaries do [1,2]. In this study, the measurements of the polycrystalline Au are used as reference measurements which could contain all these nonidealities.

[1]Belevtsev, B. I., Komnik, Yu. F. & Beliayev, E. Yu. Electron relaxation in disordered gold films. *Physical Review B* **58**, 8079–8086 (1998).

[2]Martinez, C. E. *et al.* Energy relaxation by hot 2D electrons in Algan/gan heterostructures: The influence of strong impurity and defect scattering. *Semiconductor Science and Technology* **21**, 1580–1583 (2006).

Revision. We revised the main text on page 5 to include the discussion of the effect of lattice defects and grain boundaries on hot electron dynamics in polycrystalline films.

9. The curve data in Figure 2 c and e are desultory, and not smooth enough.

Answer. We are very sorry that Reviewer had this impression. We assume that the reason for this comment is the presence of ‘sharp’ corners in $\Delta R/R$ dependence presented in these figures. We would like to point out that these features are present in both experimental and simulated curves and are true representation of the instantaneous change in the electron distribution right after the absorption of pump and then a sudden relaxation of the nonthermalized electrons. It is more obvious in Fig. 2b, when we look at $\Delta\epsilon_{2NT}$, that $\frac{d\Delta\epsilon_{2NT}}{dt}$ suddenly changes from “+” to “-” right after the pulse. On this time scale, the electrons are not thermalized yet and $\Delta R/R$ is mainly determined by the nonthermal electrons. This is the reason why both the experimental and simulated curves have sharp corners, and it is not because the time resolution in either experiments or simulations are not enough to capture the change.

Reviewer #2 (Remarks to the Author):

In this work, the thermomodulation response and interfacial transport dynamics in monocrystalline and polycrystalline Au thin films are investigated by transient reflection measurement and the three-process model based on Boltzmann-heat equations. The author shows that the band renormalization is induced by hot carriers, and that the boundary crystallinity affects the relative contribution of thermalized and nonthermalized electrons to the transient optical properties. The comparison of SiO₂ and TiO₂ substrates shows that there is hot-electron transfer from metal to substrate. The experimental and theoretical studies of non-equilibrium electron and phonon dynamics related to metal band structure, collision rates, and pump fluence have already been extensively reported in, such as ACS Photonics, 2016, 3(9):1637-1646; Light: Science & Applications, 2019, 8(1): 89; Phys. Rev. B 51, 11433 (1995); Phys. Rev. B 61, 16956 (2000); Nat. Comm. 5, 5788 (2014); Eur. Phys. J.D. 45, 369 (2007). Compared with previous works, the author mainly studied the effect of the dynamic band renormalization and electron scattering in non-equilibrium state induced by high fluence excitation. The originality and novelty it is not convincing of a significant advancement. I don’t think that the paper meets the standards of importance and novelty required by Nature Communications, and recommend to transfer the work to more focused journal.

Answer. We are very sorry to hear this opinion, which may be due to unclear presentation of our results. Entirely our fault. We completely agree that there are numerous works in which dynamics of hot electrons was studied in polycrystalline Au films. The goal of this paper is not extension of this studies to the high fluence excitation case, as Reviewer understood, but investigation of the electron dynamics in MONOCRYSTALLINE Au films.

The observed electron dynamic changes are NOT just incremental improvements which one can expect with the reduction of loss in monocrystalline materials, but reveal fundamental

distinctions with the dynamics in polycrystalline films, because of the enabled interplay between intra- and interband relaxation channels.

Our results demonstrate several features of hot-electron dynamics specific to monocrystalline ultrathin films:

- 1) a decrease in electron scattering rate and electron-phonon relaxation time in monocrystalline gold, which indicates that grain boundaries play a role in this process;
- 2) the dynamic renormalization of the interband absorption peak at the X-symmetry point of the Brillouin zone;
- 3) a strong contribution of hot-electron scattering on polar phonons in the substrate, caused by the electron spill-out and manifested by the dependence of the relaxation rate on the thermal conductivity of the substrate and thickness of gold crystals, showing that the electron spill-out plays significant but different role at different electron temperatures in electron relaxation in ultrathin monocrystalline gold;
- 4) the hot-electron injection efficiency from a monocrystalline ultrathin Au film into a semiconductor as high as ~9%, the same order of magnitude as the theoretical limit, despite the Au surface is atomically flat.

We hope that this clarifies novelty and originality of our results.

Revision. We have checked and revised the introduction and main text to make sure that the objectives of the paper and the novelty is appropriately highlighted, focusing on monocrystalline Au properties.

Specific comments can be found below.

1、 Fig. 2a shows that monocrystalline Au thin films consistently exhibit a longer rising time than polycrystalline Au films with high pump fluence. Nanoparticles with different diameters show similar hot electron behavior and electron-phonon interaction, which is caused by the weak coupling between the free electron gas and the surface electron state, and the coupling between the hot electrons and the surface accounts for less than 10% of the total electron energy losses for these particles. (The Journal of Chemical Physics 112.13 (2000): 5942-5947; ACS Photonics, 2016, 3(9): 1637-1646.) For the Au nanofilms with enhanced influence of surface electronic states, how about the hot carrier dynamics behavior in monocrystalline Au films with varying thicknesses?

Answer. We thank the reviewer for pointing out the role of surface electronic states on dynamic properties of gold nanosystems, which definitely deserves careful considerations.

a) Indeed, the increase in rise-time of transient signals in nanoparticles of small sizes [1] is routinely attributed the decrease in the Coulomb screening at the surface due to the quantum spill-out effects, that are present for all crystal facets and not only (111) surface, which supports the Shockley surface states. We expect this effect to be negligible in our case due to the confinement occurring only in one dimension, as opposed to nanoparticles, and further reduced by the mixing equilibrium and non-equilibrium dynamics at short time delays, as discussed on p. 5 of the original manuscript.

b) The first paper mentioned by Reviewer (The Journal of Chemical Physics 112.13 (2000): 5942-5947), discusses coupling of surface **acoustic modes**, as a decay channel for hot carrier cooling rather than surface electronic state. This mechanism is irrelevant to the discussion of

dynamics on sub-picosecond timescales as hot-electron cooling (with or without the contribution from a surface) takes place on picosecond timescales. However, even in this respect authors of the said paper conclude that: *“The results show that—to within our signal-to-noise — the characteristic time-scale for electron-phonon coupling does not change with the particle size for Au. This suggests that there is no effect from surface scattering on the electronic relaxation: phonon emission “inside” the particle dominates the dynamics.”* The nanoparticles studied in the above paper were between 2 and 8 nm in size with much higher surface-to-bulk ratio than our 10-15 nm thick single crystal films, therefore, in our case this surface effect is even more suppressed.

c) We are not aware of any paper to date that explicitly linked the 2D Shockley surface state on the (111) Au surfaces to observable dynamical properties in optical pump-probe experiments. We agree with Reviewer that the possibility of such connection sounds intriguing, however we would like to point out that our experiments were performed at ambient conditions, and it is likely that the surface is passivated by molecular contaminants. The results of angle-resolved photoemission spectroscopy (ARPES) and scanning tunnelling spectroscopy (STS) on gold surfaces clearly suggest that exposure of clean surfaces of gold to the environment has strong detrimental effect on the surface states [2]. Moreover time-resolved 2-photon photoemission (2PPM) and STS studies of lifetimes of electrons on surfaces on noble metals suggests the lifetimes of comparable order as bulk conduction electrons [3,4].

The second paper suggested by the Reviewer (ACS Photonics, 2016, 3(9): 1637-1646.), argues that the contribution of surface states should be comparatively small even for nanoparticles in the order of 2-6 nm in size *“... because surface electrons only make a small fraction of the total conduction electrons for the particle sizes under consideration.”*

[1] C. Voisin, D. Christofilos, N. Del Fatti, F. Vallée, B. Prével, E. Cottancin, J. Lermé, M. Pellarin, and M. Broyer; Size-Dependent Electron-Electron Interactions in Metal Nanoparticles. Phys.Rev Lett., 4 September 2000; **85**: 2200

[2] F. Forster, A. Bendounan, F. Reinert, V.G. Grigoryan and M. Springborg; S The Shockley-type surface state on Ar covered Au(111): High resolution photoemission results and the description by slab-layer DFT calculations. Surface Science, 1 December 2007; **601**(23): 5595-5604

[3] L. B'urgi, O. Jeandupeux, H. Brune, and K. S. Kern; Probing Hot-Electron Dynamics at Surfaces with a Cold Scanning Tunneling Microscope. Phys. Rev. Lett., 31 May 1999; **82**(22): 4516-4519

[4] P.M. Echenique, R. Berndt, E.V. Chulkov, Th. Fauster, A. Goldmann and U. Höfer; Decay of electronic excitations at metal surfaces. Surface Science Reports. May 2004, **52**(7-8): 219-317

Revision. We have added a brief discussion of the role of surface states on electron dynamic on p. 8 of the revised manuscript and added respective references.

2、 When discussing the effect of thickness and substrate on the carrier dynamics, the author neglects the effect of surface electronic states of Au thin films. I am not convinced. As the thickness of the single crystal Au thin film gradually decreases, the body surface area ratio of the sample changes, and the influence of the electronic state on the surface of the sample increases. The surface modification effect of the substrate on the film should be considered.

Answer. As explained above in response to the comment 1, we do acknowledge the possible role of surface electronic states on Au (111) surfaces. However, due to the measurements being performed in air, relatively high thickness of even the thinnest of our samples (10 nm) and absence of previous reports on role of Shockley surface state on hot carrier dynamics in

gold, we believe there is no evidence in our experimental data that requires us to consider them in our model.

Revision. We have added a brief discussion of the role of surface states on electron dynamic on p. 8 of the revised manuscript.

3、 The Supplementary Fig. 2 (page 5, line 11) and relevant description of Fig. 2b is missing or mislabeling. The relevant parameters of τ_{rise} under different conditions are not listed in the text and the supplementary materials.

Answer. We are very sorry for this typo and thank the reviewer for noticing it. We have corrected Fig. 2c and Fig. 2b and Supplementary figure captions. We have checked that all parameters are discussed.

Reviewer #3 (Remarks to the Author):

In this manuscript, the thermalization of the hot carriers and interfacial transport dynamics in monocrystalline and polycrystalline Au thin films with TiO₂&SiO₂ substrate are investigated by transient reflection measurement. The work claimed they have revealed the fundamental significantly work for the hot electron transfer between Au thin film and the TiO₂. However, I could not recommend its publication in Nature Communications for the reason below.

Answer. We are very sorry to hear this opinion. We would like to point out that the electron transfer studies from a monocrystalline Au is only one part of this work, and we did not claim a fundamental significance of this aspect. We believe the systematic study of the difference between the electron dynamics in monocrystalline and polycrystalline Au is of fundamental importance, which of course then influences the electron transfer process. We are very sorry it was not clear from the original manuscript.

Revision. We have checked and revised the introduction and main text to make sure that the objectives of the paper and the novelty is appropriately highlighted, focusing on monocrystalline Au properties.

1. The significance of this finding is open to debate, its indeed that increasing the hot electron transfer rate from Au to TiO₂ is important, however, the physical origin and model that the non-thermalized carriers in Au film that modulated by the grain boundaries could affect the hot electron transfer rate in the Au/TiO₂ is unclear and should be reconsidered.

Answer. We want to thank reviewer for this important comment and would like to point out that this work focuses on hot carrier dynamics in single crystal gold nanoflakes, and carrier injection from the said flakes through atomically smooth interface into TiO₂. Therefore, there are no grain boundaries to consider. As we know from the literature on the electron relaxation dynamics in polycrystalline Au films, the grain boundaries may play a significant role since they have an impact on hot-electron mean-free path, but these effects were extensively studied already [1,2,3]. Fortunately for us, there are no grains in monocrystalline Au films. We are sorry if it was not clear that we study an electron transfer from monocrystalline, atomically smooth Au.

We report the observation of efficient injection through an atomically smooth interface, for which the importance of surface roughness for relaxing the momentum conservation partially is also not important. The hot carrier transfer model used in the paper is to compute the hot-electron transfer efficiency under the fully momentum relaxation assumption. The computed efficiency values are the same order of magnitude as the experimentally measured ones, which brings us to conclude that it is still possible to obtain high hot-carrier transfer efficiencies even for monocrystalline, atomically flat Au thanks to the carrier momentum redistribution by scattering events in this very thin monocrystalline Au.

[1] Andrew J. Leenheer, Prineha Narang, Nathan S. Lewis, Harry A. Atwater; Solar energy conversion via hot electron internal photoemission in metallic nanostructures: Efficiency estimates. *J. Appl. Phys.* 7 April 2014; 115 (13): 134301

[2] Chawla, J. S., Gstrein, F., O'Brien, K. P., Clarke, J. S. & Gall, D. Electron scattering at surfaces and grain boundaries in Cu thin films and Wires. *Physical Review B* **84**, (2011).

[3] Feldman, B., Park, S., Haverty, M., Shankar, S. and Dunham, S.T. (2010), Simulation of grain boundary effects on electronic transport in metals, and detailed causes of scattering. *phys. stat. sol. (b)*, 247: 1791-1796.

<https://doi.org/10.1002/pssb.201046133>

Revision. We have checked that the distinction between polycrystalline and monocrystalline cases is clearly described in the revised manuscript.

2. The comparison between the monocrystalline and polycrystalline Au thin films is unconvincing, for example, the band structure of the Schottky contact of the Au/SiO₂ Au/TiO₂ interface and the roughness level at the real space could not be as simple as the author modeled, which could inevitably affect the relaxation process. (All the parameters in Fig. S4 is hard to tell). They have not presented the grain boundary or other crystalline parameters neither for the pump beam spot size compared with the Au film, why could they confirm the relaxation contribution of the polycrystalline Au?

Answer. We agree with Reviewer that realistic calculations of the Schottky barrier between gold at TiO₂ might require precise knowledge of the interface properties. We would like to note again that we only studied the injection between atomically flat single crystal Au and TiO₂; the case of polycrystalline Metal/Semiconductor interface is widely studied in the literature [1,2]. However, you can find the detailed characterisation of the Schottky barrier in our recent publication [3]. We described the choice of the model in our response to the comment 1 above. We are sorry that the parameters in Fig. S4 were not clear, we have now revised the figure caption to address this comment. Simple interface models like ours have been successfully used in the literature for studying injection into TiO₂ from plasmonic gold nanoparticles [4,5].

While we are not entirely sure about the meaning of the comment in the last sentence above, we show in Figure R1 the grain sizes, which have the average size of 50 nm, in the used PC Au film, and a single grain nature of the monocrystalline Au flake. The pump beam diameter, which is around 10 μm , is three orders larger than the average grain size in polycrystalline film and much smaller than the single grained monocrystalline Au flake which has the dimensions of approximately 50 μm . To this end, we attribute that the observed intriguing difference in relaxations between the cases is originated from the electron-grain boundary scattering which is present in polycrystalline Au film and absent in monocrystalline Au flake.

[1] Tagliabue, G., DuChene, J.S., Abdellah, M. *et al.* Ultrafast hot-hole injection modifies hot-electron dynamics in Au/p-GaN heterostructures. *Nat. Mater.* **19**, 1312–1318 (2020)

[2] Tagliabue, G., DuChene, J. S., Habib, A., Sundararaman, R. & Atwater, H. A. Hot-hole *versus* hot-electron transport at Cu/Gan heterojunction interfaces. *ACS Nano* **14**, 5788–5797 (2020).

[3] Kiani, F. *et al.* Transport and interfacial injection of D-band Hot Holes Control Plasmonic Chemistry. *ACS Energy Letters* **8**, 4242–4250 (2023).

[4] Ratchford, D. C., Dunkelberger, A. D., Vurgaftman, I., Owrutsky, J. C., & Pehrsson, P. E. (2017). Quantification of efficient plasmonic hot-electron injection in gold nanoparticle–tio2 films. *Nano Letters*, *17*(10), 6047–6055.

[5] Andrew J. Leenheer, Prineha Narang, Nathan S. Lewis, Harry A. Atwater; Solar energy conversion via hot electron internal photoemission in metallic nanostructures: Efficiency estimates. *J. Appl. Phys.* 7 April 2014; *115* (13): 134301.

Revision. We have checked and added the discussion of the distinction between polycrystalline and monocrystalline cases and revised the caption to Fig. S4. We have added the discussion of the grain sizes and the size of the pump beam in the revised manuscript on page 5 and SI-Methods and Sample Characterization.

3. As shown in Fig. 4b, the e-ph coupling process for low pump fluence increased vividly with the thickness of the sample, which is in discrepancy with the Te model. The issue is that the grain size of the Au thin film is not presented or remain unchanged with the thickness therefore the conclusion of the increasing electron transfer should be questionable.

Answer. We believe Reviewer refers to Fig. 4c here where approximately 15% increase in the hot-electron cooling time for low pump fluence. This observation, however, is not related to the crystallinity of the sample as the data depicted in this Figure is for **monocrystalline** gold only, no grains there. Likewise, the hot-electron transfer is also not relevant for these data, since they are obtained for the samples **on SiO₂ substrate**. Both facts are clearly stated in the figure caption. Our explanation of this effect therefore does not need to include the role of grain size or the increasing electron transfer and, as explained in the second paragraph on p. 7, is instead attributed to the electron spill-out effect in thin metal films.

Revision. No revisions needed.

REVIEWER COMMENTS

Reviewer #1 (Remarks to the Author):

In this revised manuscript, some improvements have been made to address my questions. However, I hesitate to recommend its publication in Nature Communications. The details reasons are listed below:

1.This manuscript argued that there exists a fundamental difference of the hot carrier's dynamics between the MC Au and PC Au and this difference are mainly due to the crystallinity of Au. However, the characterization data on the MC Au or PC Au is very less. It is known that there would be a fluctuation of the crystallinity even for different monocrystal samples. Thus, what's the value of the crystallinities of MC Au and PC Au? By the way, the AFM image is not the tool to characterize the crystallinity. It is noted that the MC Au is prepared by the wet chemistry method. Thus, is there any residue? If yes, will the residue affect your pump-probe measurement?

2.Is the transient reflectance spectra in the figure 1e simulated or just for illustrating? If it is based on simulation, what's you simulation method? If just for illustrating, please modify the word "simulated" in "The simulated $\Delta R/R$ " to avoid misunderstanding.

Reviewer #3 (Remarks to the Author):

In the revision, the authors' responses to my questions are sufficient and adequate. Now I am satisfied with the revised manuscript. Thus I recommend this version of the manuscript be accepted for publication

Reply to review comments

We are grateful to both reviewers for the support and the comments which helped to improve clarity of our work. In the following, we will address the remaining comments of Reviewer 1.

Reviewer #1 (Remarks to the Author):

1. This manuscript argued that there exists a fundamental difference of the hot carrier's dynamics between the MC Au and PC Au and this difference are mainly due to the crystallinity of Au. However, the characterization data on the MC Au or PC Au is very less. It is known that there would be a fluctuation of the crystallinity even for different monocrystal samples. Thus, what's the value of the crystallinities of MC Au and PC Au? By the way, the AFM image is not the tool to characterize the crystallinity. It is noted that the MC Au is prepared by the wet chemistry method. Thus, is there any residue? If yes, will the residue affect your pump-probe measurement?

Answer: We are grateful to the reviewer for raising these important questions. We are sorry that it was not clear from the original manuscript that we carefully considered these questions from the onset. The detailed studies of morphology of the samples were performed and below we briefly summarise the results. It is true that conventionally chemically grown gold crystals may not be purely single-crystal (containing more than one crystalline domain), which may randomly exist from particle to particle. The new method we developed provides monocrystalline Au micro-flakes of exceptional quality with high reproducibility [1]. By nature of the synthesis, the microflakes are single-crystals with very well-defined crystallographic surfaces. Similar monocrystalline gold flakes, also synthesized on a substrate with the same method we used except for the gap assistance, were previously investigated and proven to be monocrystalline [2-4]. We introduced the gap assistance [1] in order to improve the reproducibility and increase the lateral size of the flakes. We did a detailed characterization (XRD, HR-TEM, EDS, XPS, SAED) of individual Au MFs synthesised identically in our recent paper [1] and confirmed that these micro-flakes are atomically-smooth gold with {111} crystallographic basal surfaces and {110} crystallographic side facets (Figures R1-R4). These results demonstrate that the MC Au MF is a true single crystal and there are no fluctuations of crystallinity.

The AFM image of the MC Au micro-flake reported in the original manuscript (Fig. S1a) was included to show the thickness and flatness of the studied sample, and not to judge its crystallinity. We are sorry if this was not clear. On the contrary, AFM measurements are well established to assess the grain size in polycrystalline materials [5-7], and hence we use them to prove the granular nature of the sputtered Au film (Fig. S1b). Obviously, we cannot resolve the crystallographic orientation in the AFM image and are satisfied that it reveals small crystallites and domain boundaries. The resolution of optical measurements is not sufficient to investigate the effects of crystallographic orientations on hot-carrier dynamics, even though it can be an interesting study. For comparison of mono- and polycrystalline Au films, the subject of this work, the AFM characterization is sufficient to show the polycrystalline nature of PC Au film and the grain size in order to assert the role of the electron scattering on grain boundaries in hot-carrier dynamics (it is not the crystallographic orientation of crystallites in the PC film but the grain boundaries are important for our studies). As we mentioned in this manuscript (Supplementary Materials, p. 6), the illuminated area in the pump-probe measurements is much smaller than the lateral size of the MC flakes (we

therefore assume no contribution from the side facets in our measurements), while it covers multiple grains and grain boundaries of the PC film.

Figure R1 (Fig. 3 in [1]). Planar and cross-sectional TEM/EDS results of individual Au MFs grown by adding the optimized concentration of KBr and KCl. (a) Top-view SEM image of the Au MFs showing the studied regions on an individual flake. (b) Planar-view BRTEM image of the edge of a solution-grown Au MF marked with a blue rectangle in (a). The 1.9 nm thick continuous organic layer on the side facet of the flake is indicated on the image. (c) Atomic-resolved HRTEM image and (d) SAED pattern of the same region taken along the [111] zone axis. The 1.4 Å spacing between the {220} planes of fcc-Au and diffraction spots are labeled correspondingly. (e) Cross-sectional STEM image of a substrate-grown Au MF after FIB cross-sectioning at the region marked with a green dashed line in (a). The 1.5 nm thick continuous organic layer on the top basal facet of the flake is labeled on the image. Atomic-resolved STEM image of the region of the purple dashed rectangle is shown in Figure S17c. (f) Atomic-resolved HRTEM image of the flake-glass substrate interface region marked with a red dashed rectangle in (e). (g) Higher magnification HRTEM image and (h) SAED pattern of the twinned region of the flake within the red rectangle in (e) taken along the [110] zone axis. The 2.4 Å spacing between the {111} planes of fcc-Au, the {111} twin boundaries, and the mirrored diffraction spots is labeled correspondingly. (i) STEM image, (j) EDS line scans of Au, C, O, Cl, and Br elements, and (k) EDS spectrum obtained from the edge of the Au MF studied in (b, c). The insets in (j, k) show a magnified view of line scans of the elements right at the edge of the flake and the EDS spectrum at an energy range of 1.2–4 keV, respectively. Atomic percentages of Cl, Br, and Au elements are reported after subtracting the contributions of Cu, C, and O elements that mainly originate from the carbon-coated Cu grid.

Figure R2 (Fig. S17 in [1]). Cross-sectional TEM study results for a cross section of a substrate-grown Au micro-flake. Tilted-view FIB-SEM image of (a) the FIB-processed Au micro-flake, and (b) the fabricated TEM cross-section, i.e. a-SiO₂/Au(111)/Pt/C stacked structure, used in this study. Atomic-resolved STEM images of the regions of (c) the top basal plane and (d) the bottom basal plane of the Au micro-flake marked with red and purple dashed rectangles in Figure 3e, respectively. (e) HAADF image and EDS maps of Au, O, Ti, and Si elements, together with the corresponding EDS spectrum of the Au flake-glass substrate interface region. The magnified EDS spectra at energy ranges of 1.2-3.3 keV and 10.7-12.45 eV in the inset of (f) show the absence of Br-K α and Cl-K α characteristic peaks at 11.91 and 2.62 eV, respectively.

Figure R3 (Fig. S18 in [1]). XRD pattern obtained averaging the results from a large number of substrate-grown Au MFs.

Figure R4 (Fig. S19 in [1]). XPS measurement results obtained from a substrate-grown Au micro-flake before (i) and after (ii) washing treatment with ethanol and DI water. (a) Wide-scan XPS survey spectra of the pre-washed and washed samples. High-resolution XPS spectra of (b) Cl2p, (c) Br3d, (d,g) Au4f, (e,h) C1s, and (f,i) O1s core-levels of the samples. Detection of relatively strong Cl 2p and Br 3d core level signals for the prewashed sample and no detectable signal for the washed sample indicates the high efficiency of the implemented washing treatment in reducing the concentration of halide ions on the basal plane of the Au micro-flakes.

Regarding the effect of possible residues on the pump-probe measurements, we would like to emphasize that our unique on-substrate growth method results in MC Au MFs that are directly nucleated and grown on the glass substrate surface with no organic or halide ligands present at the Au-glass interface; i.e. the bottom Au{111}/glass interface is pristine. On-substrate growth is different from colloidal-growth approaches for which ligands exist on all surfaces. We confirmed this ligand-free Au/glass interface in our previous paper [1] by doing HR-TEM and EDS analysis on the cross-section of the Au MFs (Fig. R2). On the other hand, our HR-TEM and EDS results show that immediately after synthesis a < 2 nm organic-halide ad-layer is present on the top {111} basal surface (Fig. R1). The physically adsorbed

organic/halide residue is easily removed with a simple cleaning procedure as it is not strongly bonded (unlike a ligand) to the metal surface (Fig. R4). We have extensively discussed the role of the surface states on the electron dynamics in our previous response to the comments in the first round and made extensive revisions in the text. Most importantly, for the studies of the electron transfer processes, which would be indeed affected by a molecular layer on a surface, the micro-flakes are naturally transferred onto TiO₂ films, with the bottom, halide-free {111} surface of the Au flakes in the contact with TiO₂, so that we are sure that these interfaces are pristine.

Revision. Based on the above discussion, 1) we have checked that it is the role of grain boundaries and not crystallite orientation is discussed when comparing hot-carrier dynamics between MC and PC gold. 2) We have checked that AFM images are presented and discussed with a correct emphasis as described above. 3) We have also added in the Methods section the discussion of adlayers on the MC gold (in addition to the discussion of the role of the surface states already in the previous version of the manuscript), and 4) added the description of the cleaning procedure in the Methods section.

References:

- [1] Kiani, Fatemeh, and Giulia Tagliabue. "High Aspect Ratio Au Microflakes via Gap-Assisted Synthesis." *Chemistry of Materials* 34.3 (2022): 1278-1288.
- [2] Großmann, Swen, et al. "Nonclassical optical properties of mesoscopic gold." *Physical Review Letters* 122.24 (2019): 246802.
- [3] Kaltenecker, Korbinian J., et al. "Mono-crystalline gold platelets: a high-quality platform for surface plasmon polaritons." *Nanophotonics* 9.2 (2020): 509-522.
- [4] Rodríguez Echarri, Álvaro, et al. "Nonlinear Photoluminescence in Gold Thin Films." *ACS Photonics* (2023).
- [5] Huang, Jer-Shing, et al. "Atomically flat single-crystalline gold nanostructures for plasmonic nanocircuitry." *Nature communications* 1.1 (2010): 150.
- [6] Kevin M. McPeak, Sriharsha V. Jayanti, Stephan J. P. Kress, Stefan Meyer, Stelio Iotti, Aurelio Rossinelli, and David J. Norris *ACS Photonics* **2015** 2 (3), 326-333
- [7] Duman, A. N. (2021). Grain Analysis of atomic force microscopy images via persistent homology. *Ultramicroscopy*, 220, 113176. <https://doi.org/10.1016/j.ultramic.2020.113176>

2. Is the transient reflectance spectra in the figure 1e simulated or just for illustrating? If it is based on simulation, what's your simulation method? If just for illustrating, please modify the word "simulated" in "The simulated $\Delta R/R$ " to avoid misunderstanding.

Answer: We thank the reviewer for this comment. The transient reflection curves in Figure 1e are **the simulated $\Delta R/R$** for illustration of the mechanisms the contributions of the introduced excitation/relaxation processes. The exhaustive details of the model we used in the simulations are described in Supplementary Information Section SI.1. Transient reflectivity simulations. In order to avoid misunderstanding, we added description how we simulated the contributions of individual components. The nonthermalised, $\Delta R_{NT}/R$, and thermalized, $\Delta R_T/R$, electron contributions are simulated by considering only $\Delta\epsilon_{NT}$ and $\Delta\epsilon_T$, respectively, while not accounting for the Drude response, $\Delta\epsilon_D$. The Drude contribution in transient reflectivity is separated into two parts as an electron temperature, $\Delta R_{D-Te}/R$ and a lattice temperature, $\Delta R_{D-Tl}/R$, induced changes in the Drude response and computed by considering only elevated T_e and T_l , respectively, while not accounting the nonthermalised, $\Delta\epsilon_{NT}$, and thermalized electrons, $\Delta\epsilon_T$, changes in the permittivity of Au.

Revision. We added a brief clarification in Fig. 1e caption.

REVIEWERS' COMMENTS

Reviewer #1 (Remarks to the Author):

The authors have addressed all my issues. I recommend publication.